# Flourishing in Healthcare Trainees: Psychological Well-Being and the Conserved Transcriptional Response to Adversity

**DOI:** 10.3390/ijerph19042255

**Published:** 2022-02-16

**Authors:** Jennifer S. Mascaro, Amanda Wallace, Brooke Hyman, Carla Haack, Cherie C. Hill, Miranda A. Moore, Maha B. Lund, Eric J. Nehl, Sharon H. Bergquist, Steve W. Cole

**Affiliations:** 1Department of Family and Preventive Medicine, Division of Preventive Medicine, Emory University, Atlanta, GA 30329, USA; miranda.moore@emory.edu; 2Rollins School of Public Health, Emory University, Atlanta, GA 30322, USA; ajwallace26@gmail.com (A.W.); enehl@emory.edu (E.J.N.); 3Department of Gynecology and Obstetrics, Emory University, Atlanta, GA 30322, USA; brooke.hyman@emory.edu (B.H.); cherie.hill@emory.edu (C.C.H.); 4Department of Surgery, Emory University, Atlanta, GA 30322, USA; chaack@emory.edu; 5Department of Medicine, Emory University, Atlanta, GA 30322, USA; shoresh@emory.edu; 6Department of Family and Preventive Medicine, Physician Assistant Program, Emory University, Atlanta, GA 30322, USA; maha.lund@emory.edu; 7Department of Psychiatry & Biobehavioral Sciences, Department of Medicine, University of California, Los Angeles, CA 90095, USA; coles@g.ucla.edu

**Keywords:** well-being, flourishing, loneliness, inflammation, resident physicians, physician assistants

## Abstract

While much attention has been paid to healthcare provider and trainee burnout, less is known about provider well-being (i.e., flourishing) or about the effects of well-being on immune function. This study examined the demographic and psycho-social correlates of well-being among healthcare trainees (resident physicians and physician assistant (PA) trainees) and evaluated the association of well-being with the “conserved transcriptional response to adversity” (CTRA) characterized by up-regulated expression of pro-inflammatory genes and down-regulated expression of innate antiviral genes. Participants (n = 58) completed self-reported assessments of sleep disturbance, loneliness, depressive symptoms, anxiety, stress, and well-being (flourishing). Blood sample RNA profiles were analyzed by RNA sequencing to assess the CTRA. Slightly over half (n = 32; 55.2%) of healthcare trainees were categorized as flourishing. Flourishing was less prevalent among primary caregivers, and more prevalent among trainees who exercised more frequently and those with fewest days sick. Loneliness (AOR = 0.75; 95% CI = 0.61, 0.91; *p* = 0.003) and stress (AOR = 0.65; 95% CI = 0.45, 0.94; *p* = 0.02) were associated with decreased odds of flourishing when controlling for other variables. Flourishing was associated with down-regulated CTRA gene expression, whereas loneliness was associated with up-regulated CTRA gene expression (both *p* < 0.05). Assessing these relationships in a larger, multi-site study is of critical importance to inform policy, curricula, and interventions to bolster sustainable trainee well-being.

## 1. Introduction

Over the last decade, extensive research and attention have highlighted the concerning prevalence of burnout (characterized as emotional exhaustion, depersonalization, and reduced personal accomplishment) and depression among healthcare providers and trainees, largely due to factors in the institutional environment that exacerbate chronic psycho-social stress, such as excessive workloads, work-life conflicts, lack of control, decreased autonomy, clerical burdens, and increased use of technology [1,2,3,4,5,6,7,8,9]. While incoming medical students report better well-being than matched controls, their levels of depression, anxiety, and distress increase throughout their training, ultimately reaching higher levels than those observed in their age-matched peers [10,11,12,13,14,15,16]. By residency, an estimated 50–75% of residents report at least one symptom of burnout, with large variability by clinical specialty [17,18,19], and over a quarter of medical residents experience depression [20]. While far less research has been conducted to examine burnout and mental health among physician assistants (PAs) and PA students, burnout appears to be highly prevalent, with estimates of 34–80% of PAs experiencing burnout [21,22]. In a recent study, almost 80% of PA students met criteria for at least one of the burnout dimensions [23]. The extraordinary levels of suffering among frontline medical providers have profoundly harmful effects on quality of care, patient satisfaction, safety, and treatment adherence [21,24,25,26,27,28,29,30,31,32,33]. In addition, provider burnout and depression increase attrition and contribute to the current and estimated shortages of physicians and PAs [13,34,35,36,37,38,39,40].

Burnout is also associated with detrimental mental and physical health outcomes among providers themselves, including elevated risk for type 2 diabetes, hypercholesterolemia, coronary heart disease, musculoskeletal pain, headaches, gastrointestinal and respiratory problems, severe injuries, and mortality below the age of 45 [41,42,43,44]. The relationship between healthcare provider well-being and health outcomes may be mediated, in part, by a stress-induced gene regulatory program known as the conserved transcriptional response to adversity (CTRA), which is characterized by the up-regulation of pro-inflammatory genes and the down-regulation of type I interferon-related innate antiviral genes in circulating immune cells [45,46]. A wide range of psychiatric sequela and adverse experiences are associated with increased inflammatory activity, including depression, anxiety-related disorders, chronic stress, and discrimination [47,48,49,50]. Similarly, a large body of research has demonstrated the association between social isolation and the CTRA profile of up-regulated pro-inflammatory gene expression and down-regulated antiviral gene expression [51,52,53,54]. Mechanistic analyses have demonstrated that harmful psycho-social conditions lead to elevated CTRA profiles via sympathetic nervous system-mediated gene regulation [55]. In contrast, people who report high levels of eudemonic meaning and individuals that engage in prosocial behavior have been found to have reduced CTRA expression profiles [56,57,58,59,60,61].

While research highlights a striking prevalence of burnout and mental health symptoms, as well as the downstream harm that these impart on both patient and provider outcomes, far less is known about provider positive well-being. According to influential theoretical models and empirical analyses of well-being, flourishing is a state of positive health that includes high levels of emotional, psychological, and social well-being [62]. This definition aligns with the concept of mental health characterized by the World Health Organization (WHO) as “a state of well-being in which the individual realizes his or her own abilities, can cope with the normal stresses of life, can work productively and fruitfully, and is able to make a contribution to his or her community” [63]. Although mental health and mental illness tend to be inversely associated, they are distinct constructs rather than simply two sides of a single coin [64]. Previous research has linked flourishing with more optimal biomarkers of inflammation [60,65], positive health behaviors and outcomes [66,67,68,69], and job satisfaction [70,71,72]; however, few studies have examined healthcare trainees. One recent study of resident physicians found that flourishing as measured by the mental health continuum (MHC) was positively associated with resilience and coping and negatively correlated with emotional exhaustion and negative affect [73]. Here, we conducted an exploratory study to assess the relationship between flourishing and mental health symptoms, psycho-social characteristics, and CTRA gene regulation among healthcare trainees.

## 2. Materials and Methods

Study overview: This cross-sectional study was conducted using baseline data collected as part of a larger study examining the impact of app-delivered mindfulness for healthcare trainee well-being (NCT03452670; the results of the randomized, longitudinal study are presented elsewhere). We recruited four different groups of healthcare trainees for participation in this study from March 2018 to April 2020: medical residents from three specialties (family medicine (FM), obstetrics/gynecology (OB/GYN), and general surgery) and PA students. Here, we examine the baseline associations among well-being, mental health symptoms, psycho-social characteristics, and CTRA gene expression prior to randomization and allocation in the interventional aspect of the larger study. The study was approved by the Emory Institutional Review Board (IRB). All participants were recruited at Emory University and informed of the study’s purpose, assessment procedures, confidentiality, compensation, potential risks and benefits, and voluntary nature of participation, including the right to stop participating at any time without penalty. Upon providing informed consent in accordance with the Emory IRB standards, participants were assessed for baseline levels of depressive symptoms, anxiety, stress, sleep disturbance, loneliness, and well-being through a series of self-report questionnaires administered through Qualtrics (Provo, UT). Next, we acquired blood samples from 3 of the 4 trainee groups. The FM residents did not provide blood samples due to financial and scheduling constraints.

Participants: Inclusion criteria included current Emory University medical residents in the general surgery, OB/GYN, or FM programs and current Emory University PA students. There were no exclusion criteria for cohorts.

Measures: Demographics: Self-reported demographic information included sex, age, relationship status (single, in a relationship, married), number of children, race, ethnicity, number of days sick in the previous 30 days, and number of times exercised in the previous 30 days.

Sleep Disturbance: Sleep disturbance was assessed using the patient-reported outcomes measurement information system (PROMIS) sleep disturbance short form [74], an 8-item scale with answer options ranging from (1) not at all, never, or very poor to (5) very much, always, or very good. The survey asked participants to indicate how often each item described them over the past 7 days. Sample items include: “My sleep was restless” and “I had trouble staying asleep”. Scores could range from 8 to 40, with higher scores indicating more severe levels of sleep disturbance. To categorize levels of sleep disturbance, scores were converted into T-scores ranging from 28.9 to 76.5. T-scores of less than 55 indicate none to slight levels, scores between 55.0–59.9 indicate mild levels, scores between 60.0–69.9 indicate moderate levels, and scores over 70 indicate severe levels of sleep disturbance [75]. The Cronbach alpha reliability for this scale was 0.87, suggesting high internal consistency of scale items.

Loneliness: Loneliness was assessed using the revised UCLA loneliness scale [76], a 20-item scale with answer options ranging from (1) never to (4) often. The survey asked the participant to indicate how often each item described them. Sample items include “My social relationships are superficial” and “My interests and ideas are not shared by those around me”. Scores could range from 20 to 80, with higher scores indicating higher perceived levels of loneliness. The Cronbach alpha reliability for this scale was 0.92, suggesting high internal consistency of scale items.

Depression, Anxiety, Stress: Depression, anxiety, and stress were assessed using the depression, anxiety, and stress scale (DASS21) short form [77], a 21-item scale with answer options ranging from (0) “Did not apply to me at all” to (3) “Applied to me very much, or most of the time”. The survey asked the participant to indicate how often each item described them over the past 7 days and scores for each of the three subscales can range from 0–21.

The depression scale (e.g., “I couldn’t seem to experience any positive feeling at all”) demonstrated a Cronbach alpha reliability of 0.85, suggesting high internal consistency of scale items. To categorize levels of depression, scores of 0–4 indicate normal levels, scores of 5–6 indicate mild levels, scores of 7–10 indicate moderate levels, scores of 11–13 indicate severe levels, and scores greater than 14 indicate extremely severe levels.

The anxiety scale, e.g., “I experienced breathing difficulty (e.g., excessively rapid breathing, breathlessness in the absence of physical exertion)”, demonstrated a Cronbach alpha reliability of 0.58, suggesting that the internal consistency of scale items is minimal. To categorize levels of anxiety, scores of 0–3 indicate normal levels, scores of 4–5 indicate mild levels, scores of 6–7 indicate moderate levels, scores of 8–9 indicate severe levels, and scores greater than 10 indicate extremely severe levels.

The stress scale, e.g., “I was intolerant of anything that kept me from getting on with what I was doing”, demonstrated a Cronbach alpha reliability of 0.75, suggesting moderate internal consistency of scale items. To categorize levels of stress, scores of 0–7 indicate normal levels, scores of 8–9 indicate mild levels, scores of 10–12 indicate moderate levels, scores of 13–16 indicate severe levels, and scores greater than 17 indicate extremely severe levels.

Well-Being: Well-being was assessed using the adult mental health continuum short form (MHC-SF) [78], a 14-item scale with answer options ranging from (0) never to (5) every day. The survey asked the participant to indicate how often they experienced or felt each item over the past month. The scale has two subscales: hedonic well-being and eudemonic well-being. Hedonic well-being consists of one subscale: emotional well-being. Scores could range from 0 to 15, with higher scores indicating increased levels of emotional well-being. The Cronbach alpha reliability for the emotional well-being scale was 0.87, suggesting high internal consistency of scale items.

Eudemonic well-being consists of two scales: social well-being and psychological well-being. The social well-being scale included items “That you had something important to contribute to society” and “That you belonged to a community (like a social group, or your neighborhood)”. Scores could range from 0 to 25, with higher scores indicating increased levels of social well-being. The Cronbach alpha reliability for the social well-being scale was 0.82, suggesting high internal consistency of scale items. The psychological well-being scale included items “Good at managing the responsibilities of your daily life” and “That you had experiences that challenged you to grow and become a better person”. Scores could range from 0 to 30, with higher scores indicating increased levels of psychological well-being. The Cronbach alpha reliability for the stress scale was 0.87, suggesting high internal consistency of scale items.

We used the MHC-SF to categorize participants’ mental health as flourishing, languishing, or moderate according to standard and validated scoring instructions [78]. Participants were categorized as flourishing if they reported at least 1 item of hedonic well-being and at least 6 items of eudemonic well-being as (4) almost every day or (5) every day. Participants were categorized as languishing if they reported at least 1 item of hedonic well-being and at least 6 items of eudemonic well-being as (0) never or (1) once or twice. Participants who did not meet either of these criteria were categorized as having moderate mental health.

CTRA Gene Expression: The CTRA gene expression profile was quantified using a standard RNA composite analysis involving pre-specified sets of pro-inflammatory and type I interferon innate antiviral response genes, as in previous research [45,46]. Whole blood samples were collected by two different methods (PAXgene RNA tubes, and dried blood spot (DBS)) and shipped on dry ice to the University of California, Los Angeles Social Genomics Core Laboratory for RNA profiling. Total RNA from peripheral blood mononuclear cells (PBMCs) was obtained through an automated nucleic acid extraction system (Qiagen QIAcube; Qiagen Inc., Valencia, CA, USA), using the manufacturer’s standard protocol and reagents (RNeasy Mini; Qiagen Inc., Valencia, CA, USA). It was then tested for suitable mass and integrity (Thermo-Fisher NanodropOne and Agilent 2200 TapeStation; Agilent, Santa Clara, CA, USA). Total RNA from DBS was extracted by cutting blood spots from filters, micro-centrifuging them, and obtaining them through the same automated nucleic acid extraction system (Qiagen QIAcube; Qiagen Inc., Valencia, CA, USA), using the manufacturer’s standard protocol and reagents (RNeasy Micro; Qiagen Inc., Valencia, CA, USA). Mass and integrity were not assessed in DBS samples as the RNA quantity falls below the lower limit of detection for currently available systems. RNA from both methods was converted to cDNA libraries using a high-efficiency mRNA-targeted reverse transcription system (Lexogen QuantSeq 3′ FWD; Lexogen Inc., Greenland, NH, USA) and sequenced on an Illumina NextSeq 500 instrument (Lexogen GmbH; Vienna, Austria), following the manufacturers’ standard protocols for this workflow. Assays targeted 5 million reads/sample (an average of 4.9 million was achieved), each of which was mapped to the reference human transcriptome (hg38) using the STAR aligner and quantified as gene transcripts per million mapped reads. Gene expression values were log2-transformed and mean-centered by gene for mixed effect linear model analyses as validated in previous research [79]. Analyses used mixed effect linear models to test the association between flourishing (or other variables as indicated) and the average expression of 43 z-score standardized CTRA indicator genes (from a list of 53 previously employed, after removal of 6 antiviral and 4 pro-inflammatory gene transcripts that showed minimal levels and variation in this dataset (SD < 0.5 log2 expression)). The 28 type I interferon-related transcripts were sign-inverted, such that their contributions to the CTRA profile were inverse.

Data Analysis Description: We recruited a convenience sample of all interested members of each of the three residency programs and of PA students; thus, the sample size was based on the limits of participant availability. As a statistical power analysis would not have affected the study design, we did not conduct a power analysis prior to the study, and we enrolled all available consenting participants from each of the trainee groups. All analyses were conducted with Statistical Package for the Social Sciences software (SPSS), version 27.0 (IBM, Armonk, NY, USA) and SAS Version 9.4 (SAS Institute Inc., Cary, NC, USA). Statistical significance was evaluated at the 0.05 level. First, descriptive statistical analyses were conducted to characterize the baseline sample of healthcare trainees. Missing items within scales were accounted for using expectation maximization [80] (missing items never accounted for more than 5% of the total data). To evaluate normality, we assessed skewness and kurtosis. The depression scale had skewness values of <2. All other scales and subscales had skewness values of >1. As all skewness values were <3.29, these values were judged to be adequate [81]. The depression scale had a kurtosis value of approximately 3 and the emotional well-being subscale had a kurtosis value of <2. All other scales and subscales had kurtosis values of >1. As all kurtosis values were <3.29, these values were judged to be adequate [81].

To examine the bivariate associations between participant characteristics, mental health symptoms, psycho-social characteristics, and well-being, Pearson’s R and Spearman’s Rho correlations were used based upon the level of data. We used multi-variate logistic regression analyses to explore the associations among participant characteristics, mental health symptoms, psycho-social characteristics, and well-being for statistically significant inputs. Analyses were also conducted among medical resident clinical specialties, between all medical residents and PA students, and across the total sample to examine if differences in participant characteristics, mental health symptoms, psycho-social characteristics, and well-being were statistically significant. Preliminary analyses for model variable selection consisted of bivariate logistic regression for all participant characteristics, mental health symptoms, and psycho-social characteristics as the independent variable with the binary dependent variable of well-being (flourishing equals 1). Criteria for variable selection in the models were based on a bivariate statistically significant association, where *p* < 0.05. We assessed multi-collinearity as part of the regression analyses and all VIFs were <3, indicating no critical levels of multi-collinearity were present.

To identify the correlates of well-being in healthcare trainees, we used sequential multi-variate logistic regression to examine the relationship between MHC-SF-defined flourishing and mental health symptom and psycho-social characteristic inputs. First, we conducted bivariate logistic regression analyses to identify statistically significant demographics and inputs with flourishing. Next, statistically significant demographic characteristics were included in step 1, health behaviors in step 2, and mental health symptom and psycho-social characteristic inputs in step 3 of the logistic regression model. The adjusted odds ratio and *p*-value were calculated for the association between each input and the dependent variable of flourishing. The Nagelkerke R^2^ value was calculated and compared between each step to evaluate the goodness-of-fit of the regression model. For analyses of CTRA gene expression, SAS PROC MIXED (SAS Version 9.4, SAS Institute Inc., Cary, NC, USA).was used to fit a mixed effect linear model relating average expression of the 43 CTRA indicator genes to specified variables (e.g., flourishing), while controlling for blood collection modality (dried blood spots vs. PAXgene RNA tubes), a random subject-specific intercept term to control for repeated measures (indicator genes), and other covariates as noted.

## 3. Results

### 3.1. Participant Characteristics

Fifty-nine trainees agreed to participate in the study: general surgery residents (n = 11), OB/GYN residents (n = 12), FM residents (n = 21), and PA students (n = 15) (Figure 1, Table 1). One FM resident did not complete the self-report battery and only provided demographic data. Thus, the sample size for the self-report measures was N = 58. To examine differences between residents (n = 44, 74.6%) and PA students (n = 15, 25.4%), independent samples t-tests and Pearson’s chi-squared tests of independence were performed. There were no statistically significant differences among the personal demographic characteristics between trainee types. A series of one-way ANOVAs and Pearson’s chi-squared tests of independence were conducted among each of the specialty types (N = 59: n = 11 general surgery residents; n = 12 OB/GYN residents; n = 21 FM residents; n = 15 PA students). There were no statistically significant differences between the groups in terms of personal demographic information.

### 3.2. Mental Health Symptoms, Psycho-Social Characteristics, and Well-Being

A total of 17 (29.3%) participants indicated sleep disturbance symptoms, 18 (31.0%) participants indicated symptoms of depression, 26 (44.8%) participants indicated symptoms of anxiety, and 18 (31.0%) participants indicated symptoms of stress (Table 2). Well-being measures identified 32 (55.2%) participants as flourishing and 26 (44.8%) as non-flourishing, with one (1.7%) categorized as languishing and 25 (43.1%) categorized as having moderate well-being.

Depressive symptoms were significantly higher among residents (mean = 3.86; SD = 3.54) than PA students (mean = 2.20; SD = 1.57; t = −2.46, df = 52.59, *p* = 0.017). Among residents, 37.2% (n = 16) were categorized as having depressive symptoms as compared to 13.3% (n = 2) of PA students. No significant differences were observed by trainee type or among each of the specialties for any other scores. Correlation analyses also found no significant associations between the sleep (*p* = 0.29), depression (*p* = 0.09), anxiety (*p* = 0.17), stress (*p* = 0.09), and mental health (*p* = 0.66) categorizations with trainee type (Table 2). Additionally, no significant associations were found between the sleep (*p* = 0.36), depression, (*p* = 0.22), anxiety (*p* = 0.22), stress (*p* = 0.37), and mental health (*p* = 0.40) categorizations with specialty type (results not shown).

### 3.3. Bivariate Analyses Regarding Participant Characteristics and Mental Health Symptoms, Psycho-Social Characteristics, and Well-Being

As shown in Table 3, correlation analyses linked relationship status to higher levels of stress (r = 0.31, *p* = 0.02), and primary caregiver status to higher levels of loneliness (r = 0.26, *p* = 0.05) and depression (r = 0.42, *p* = 0.001). Number of days sick in the previous 30 days was associated with higher levels of loneliness (r = 0.28, *p* = 0.04) and anxiety (r = 0.34, *p* = 0.008). Number of times exercised in the previous 30 days was associated with lower levels of loneliness (r = −0.37, *p* = 0.005), depression (r = −0.44, *p* = 0.001), and stress (r = −0.29, *p* = 0.03) and higher levels of emotional well-being (r = 0.46, *p* < *0*.001), social well-being (r = 0.29, *p* < 0.03), and psychological well-being (r = 0.32, *p* = 0.02).

As shown in Table 4, sleep disturbance was associated with anxiety (r = 0.39, *p* = 0.003), and loneliness was positively associated with depression (r = 0.57, *p* < 0.001), anxiety, (r = 0.46, *p* < 0.001), and stress (r = 0.27, *p* = 0.04) and negatively associated with emotional well-being (r = −0.59, *p* < 0.001), social well-being (r = −0.52, *p* < 0.001), and psychological well-being (r = −0.58, *p* < 0.001).

#### 3.3.1. Research Question 1: What are the Associations Between Mental Health Symptoms and Psycho-Social Characteristics with Flourishing?

Bivariate logistic regression analyses found flourishing to be inversely associated with being a primary caregiver (*p* = 0.03), number of days sick in the previous 30 days (*p* = 0.04), loneliness (*p* < 0.001), depression (*p* = 0.02), anxiety (*p* = 0.02), and stress (*p* = 0.01), and positively associated with the number of days exercised in the previous 30 days (*p* = 0.05; results not shown). Flourishing rates did not differ as a function of age (*p* = *0*.69), gender (*p* = 0.08), relationship status (*p* > 0.05 for all categories), race (*p* > 0.05 for all categories), ethnicity (*p* = 0.64), trainee specialty (*p* > 0.05 for all categories), or sleep (*p* = 0.32).

In the first step of the multi-variate logistic regression model, primary caregivers were significantly less likely to be flourishing as compared to those who were not primary caregivers (AOR = 0.09; 95% CI = 0.01, 0.78; *p* = 0.03; Table 5). Trainee type did not significantly predict flourishing (*p* = 0.90). In the second step, none of the variables significantly predicted flourishing. In the third step, each unit increase in loneliness (AOR = 0.75; 95% CI = 0.61, 0.91; *p* = 0.003) and stress (AOR = 0.65; 95% CI = 0.45, 0.94; *p* = 0.02) was associated with decreased odds of flourishing. Trainee type (*p* = 0.55), being a primary caregiver (*p* = 0.08), number of days sick in the previous 30 days (*p* = 0.63), number of days exercised in the previous 30 days (*p* = 0.66), anxiety (*p* = 0.38), and depression (*p* = 0.06) did not significantly predict flourishing when controlling for the other variables. Psycho-social variables significantly enhanced the prediction of flourishing, as the Nagelkerke R^2^ increased from 0.24 in step two to 0.64 in step three. The Hosmer–Lemeshow test for the third step was not significant (X^2^ = 8.79; df = 8; *p* = 0.36) indicating goodness-of-fit for this model. The VIF for each variable was less than 2.5.

#### 3.3.2. Research Question 2: Are Mental Health Symptoms, Psycho-Social Characteristics, or Flourishing Associated with Pro-Inflammatory Gene Expression?

Mixed linear model analyses identified significantly reduced CTRA gene expression among flourishing trainees (−0.173 log2 mRNA abundance ±0.072 SE, *p* = 0.02; Table 6), and similar results emerged after additional control for age, sex, race, current illness symptoms, history of heavy alcohol consumption, and trainee type (−0.174 ± 0.087, *p* = 0.06). Additional exploratory analysis of other measures of mental health and illness (e.g., depression, anxiety, etc.) found that only loneliness showed a significant association with CTRA gene expression (0.008 log2 mRNA abundance per SD loneliness, ±0.003 SE, *p* = 0.02), and this effect also remained significant despite control for the covariates listed above (0.011 ± 0.005 SE, *p* = 0.03). Follow-up analyses that included both flourishing and loneliness in the same model failed to find a significant association for either variable (both *p* > 0.16), suggesting that CTRA gene expression is most directly associated with the variance shared by flourishing and social integration (i.e., low loneliness).

## 4. Discussion

Given the high prevalence of burnout among healthcare providers, understanding the sources of well-being among healthcare trainees in the earliest stages of their career is of the utmost importance to inform targeted approaches to prevent and mitigate burnout and maintain resilience in healthcare professionals. Identifying approaches to reduce burnout and bolster well-being have proven challenging [82,83], and the lack of studies on flourishing among healthcare trainees is a critical impediment.

Overall, the rates of flourishing among trainees in our study (55% of all trainees, 53.5% of residents, and 60% of PA students) are higher than rates found in the general population. For example, an early study of over 3000 adults in the United States between the ages of 25 and 74 revealed that only 17% fit the criteria for flourishing, with 12% of adults meeting the criteria for languishing [62]. Lebares and colleagues’ longitudinal survey of mixed-specialty first year trainees found rates of flourishing between 76% to almost 86%, depending on when the resident was surveyed [73]. In combination, the findings from our study and those from the study of Lebares et al. indicate that flourishing remains high among resident physicians, even while burnout and depression rates are of concern. If this general finding holds, it is important for understanding the overall health and well-being of healthcare trainees and it further underscores that, although positive mental health and symptoms of mental illness are often inversely related, they are distinctive factors that should be carefully measured and disambiguated [64,84].

Loneliness and stress remained independent negative predictors of flourishing when included in the same multi-variate analysis. Previous studies examining flourishing (as measured by the MHC) among residents found that flourishing was positively associated with workplace support, but not with workplace demand [73,85]. Together, these findings further highlight the importance of social connection for well-being among resident physicians. While far less is known about PA student well-being, our findings indicate that similar mental health inputs are associated with PA student flourishing.

We also identified socio-demographic factors associated with flourishing. There was no statistically significant difference in rates of flourishing or well-being scores between trainee or resident specialty type. Healthcare trainees who were also primary caregivers were less likely to meet criteria for flourishing, although the association was no longer significant when controlling for other inputs such as stress and loneliness. Previous research noted the difficulties encountered by parents in residency programs during pregnancy and after childbirth [86,87]. While our findings indicate that trainees with caregiving responsibilities may struggle due to the accompanying stress and feelings of disconnection that arise for parents in an already stressful environment, this study was not designed or powered to identify factors mediating the effect of caregiving on well-being, and future larger studies should investigate this further. A growing body of research highlights the importance of work–life integration, operationalized as the extent to which a provider feels compelled to choose between multiple competing interests (e.g., sleeping, eating, spending time with family) given limited time and/or resources [88]. Supporting the work–life integration of trainees and providers with policies and programs to increase autonomy, flexibility, and support will be crucial for flourishing, especially among those with caregiving demands.

Significantly more residents (37%) were categorized with depressive symptoms than PA students (13%). Rates of depressive symptoms in our study were approximately 10% higher than previous studies examining depressive symptoms in residents [20,89]. While depression remains under-researched among PA students, the 13% of PA students reporting depressive symptoms in our study was higher than the 8% positively screened for depression in a previous study [90]. Additionally, another study found almost 80% of PA students reported burnout in the exhaustion dimension with a similar amount expressing interest in participating in a program to reduce burnout and improve their well-being [23]. More research on PA burnout, depression, and well-being is crucially needed, especially as PAs and PA students may have unique needs and circumstances. The PA profession is a growing workforce that is increasingly young and female [22,91], both demographic categories that have increased risk for burnout and depression [5,92,93]. Our study found that rates of flourishing did not differ between PA students and residents, and that, for the most part, PA students had similar profiles of sleep disturbance, anxiety, stress, loneliness, and well-being as resident physicians.

A growing body of research is focused on identifying evidence-based approaches to improving psycho-social flourishing [94]. Our findings highlight behavioral factors to consider when creating environments conducive to trainee flourishing. Trainee exercise frequency was also positively associated with well-being and lower levels of loneliness, depression, and stress. While this is not surprising given the many studies linking exercise with well-being [95], rates of exercise among the trainees in our study were remarkably low. Trainees reported exercising on average fewer than seven times per month, with residents exercising fewer than six times per month. A previous study that compared faculty physicians and residents on a number of quality-of-life measures found that residents exercise less than faculty, with 41% of residents reporting that they exercised three to five times per week as compared to 54% of faculty physicians [95]. Identifying ways to facilitate exercise among healthcare trainees is a promising approach to increase well-being and prevent deleterious mental health outcomes [96,97].

In line with prior studies of sick leave in nurses, we found the number of sick days was positively associated with higher levels of loneliness and anxiety [98]. Interestingly, while almost a third of the trainees in this study reported sleep disturbance, levels of sleep disturbance were not significantly associated with flourishing or with other symptoms of mental illness. The one exception was anxiety, which was highly associated with levels of sleep disturbance. Previous research indicates that residents and other trainees often experience high rates of sleep dysfunction [99], and the resulting fatigue has profoundly harmful effects on cognition, patient safety, and trainee health [100,101,102]. While we did not find such an association between sleep dysfunction and poor outcomes in our study, the relatively small sample size and the cross-sectional nature of the analyses may have limited our power to detect this relationship.

We also found that flourishing was related to a molecular indicator of biological well-being—reduced CTRA gene expression—whereas loneliness was associated with elevated CTRA gene expression. Although a causal relationship cannot be definitively drawn from these findings, mental health symptoms and psycho-social behaviors are interrelated with physical health and are important factors in understanding physician and trainee well-being. Research links eudemonic well-being with favorable health outcomes [60,103,104,105,106,107,108], an association that may be mediated, at least in part, by the reduction in pro-inflammatory signaling that accompanies states of psycho-social flourishing [60,61]. We found the same association among trainees, with flourishing trainees demonstrating reduced CTRA gene expression. Our results are consistent with previous studies showing CTRA gene expression to be down-regulated in association with flourishing and up-regulated in association with loneliness and isolation [60,109]. Our findings also indicate that CTRA gene expression is most directly associated with the variance shared by flourishing and social connection, in as much as neither variable was significantly associated with flourishing when entered in the same model. Previous randomized control experiments identified interventions that reduce CTRA gene expression and improve psycho-social distress, including mindfulness-based interventions [110,111]. Future research to examine the impact of policies and interventions on trainee well-being should include biological measures that can more definitively evaluate whether improved flourishing alters CTRA and the associated health outcomes, and the results from this study point to social connection being among the most important factors to consider in such efforts. Moreover, well-powered prospective studies are crucial to more definitively establish the protective and risk factors for trainee flourishing, and to identify the individual, dyadic, communal, and system-level mechanisms by which social connection and support lead to flourishing.

Although this study had several strengths, including being among the first to examine resident physician and PA student flourishing and its relationships with inflammatory processing, several limitations warrant further discussion. First, our sample size is small, both within each cohort and across the entire study. Our study may have been underpowered to detect smaller, but important, associations between socio-demographic and behavioral factors, well-being, and CTRA gene expression. Our sample is also from a single institution, which limits generalizability. These data were collected from the subset of trainees who chose to participate in a larger study that evaluated a mindfulness intervention, and it is possible that self-selection biases influenced our dataset such that the symptoms, well-being, and physiology of our sample is not representative of the larger population. Here, we used the MHC-SF to quantify flourishing, but other flourishing measures have been used in the context of physician residents [112]. Interestingly, to our knowledge, the only other study to examine flourishing using an alternative measure found results that are discrepant with those found here and with the other studies that used the MHC-SF. Specifically, this study and the previous two that used the MHC-SF found higher rates of flourishing among residents than has been found in a general adult population [73,85]. However, the study that used VanderWeele’s flourishing measure found lower flourishing scores, prompting the interpretation that medical residents have *lower* rates of flourishing than community samples [112]. More research using multiple measures is crucial to characterize rates of flourishing among healthcare trainees and providers, to identify factors associated with flourishing, and to evaluate the long-term impact of flourishing on provider health and performance outcomes.

## 5. Conclusions

Incorporating positive psychological measurements and approaches into healthcare trainee programs is a common mandate and point of emphasis [85,112,113,114,115], and our findings point to socio-demographic factors and behavioral associations that can help identify and refine impactful and feasible programming to bolster healthcare trainee flourishing. Our findings indicate that facilitating the behaviors associated with flourishing is especially critical among primary caregivers to improve well-being and reduce loneliness and stress. Ultimately, our results highlight the importance of curricula, workplace systems, resources, and cultures that facilitate social connections and support and that make healthy lifestyles more feasible within healthcare training programs and organizations.

## Figures and Tables

**Figure 1 ijerph-19-02255-f001:**
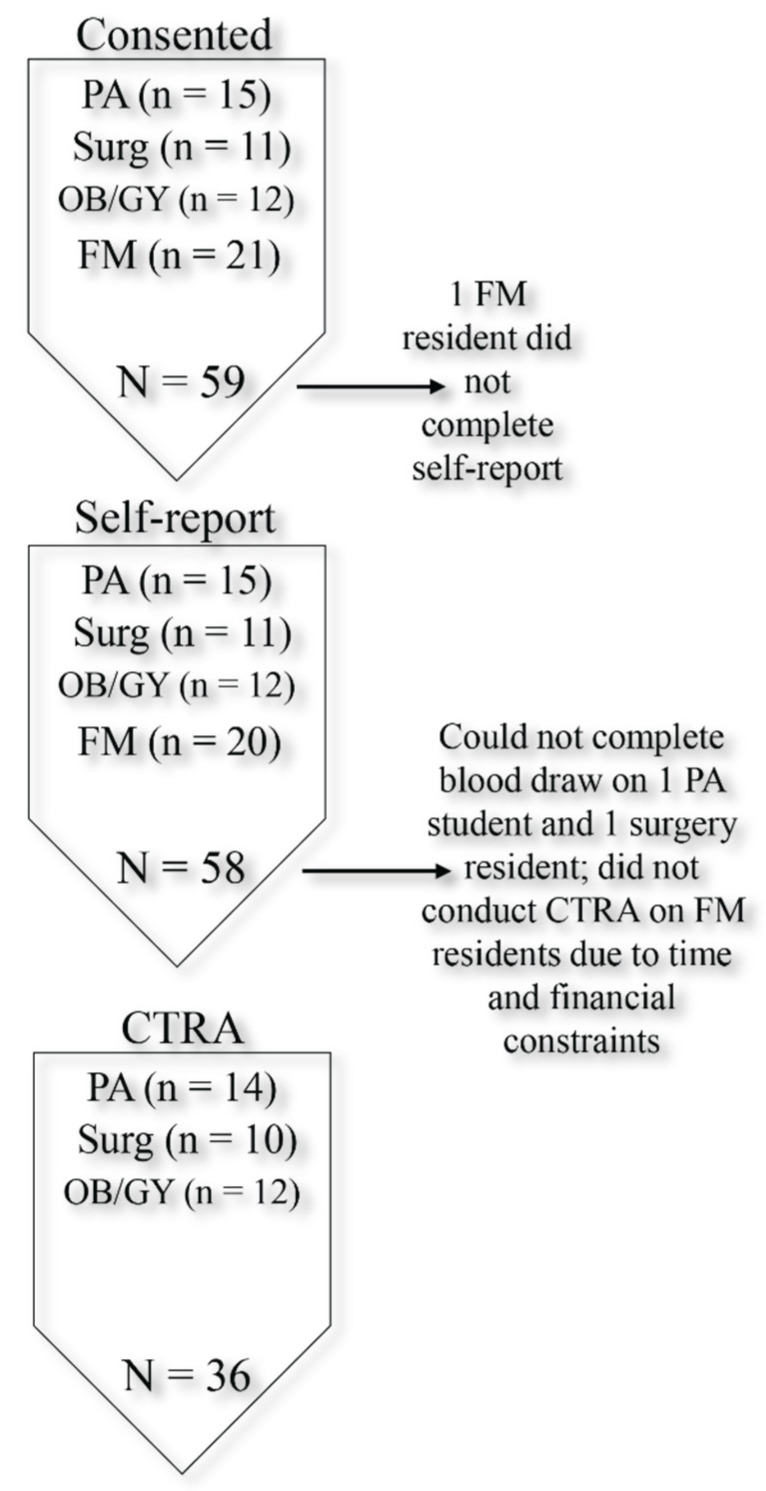
Participant flow chart.

**Table 1 ijerph-19-02255-t001:** Participant characteristics.

	Total Sample (n = 59)	Residents (n = 44)	PA Students (n = 15)	Test Statistic (Degrees of Freedom), *p*-Value
Age	27.8 (SD = 3.03)	30.13 (SD = 2.85)	28.74 (SD = 3.42)	T(df = 57) = −1.55, *p* = 0.13
Sex				X^2^(df = 1) = 0.07, *p* = 0.80
Female	45 (76.3%)	33 (75.0%)	12 (80.0%)	
Male	13 (22.0%)	10 (22.7%)	3 (20.0%) *	
Nonbinary **	1 (1.7%)	1 (1.7%)	0 (0.0%)	
Relationship Status				X^2^(df = 2) = 1.39, *p* = 0.50
Single	16 (27.1%)	11 (25.0%)	5 (33.3%) *	
Relationship	24 (40.7%)	17 (38.6%)	7 (46.7%)	
Married	19 (32.2%)	16 (36.4%)	3 (20.0%) *	
Primary Caregiver	8 (13.6%)	7 (15.9%)	1 (6.7%) *	X^2^(df = 1) = 0.82, *p* = 0.37
Race				X^2^(df = 3) = 3.42, *p* = 0.33
White	37 (62.7%)	25 (56.8%)	12 (80.0%)	
AA/Black	8 (13.6%)	7 (15.9%)	1 (6.7%) *	
Asian	9 (15.3%)	7 (15.9%)	2 (13.3%) *	
Other	5 (8.5%)	5 (11.4%) *	0 (0.0%) *	
Hispanic or Latino				X^2^(df = 1) = 0.08, *p* = 0.78
Yes	3 (5.2%)	2 (4.5%) *	1 (6.7%) *	
No	54 (93.1%)	40 (90.9%)	14 (93.3%)	
Unknown **	1 (1.7%)	1 (2.3%)	0 (0.0%)	
Days Sick/month	1.06 (SD = 1.64)	1.14 (SD = 1.83)	0.80 (SD = 0.94)	T(df = 47.49) = −0.92, *p* = 0.36
Exercised/month	6.61 (SD = 6.73)	5.73 (SD = 6.41)	9.20 (SD = 7.20)	T(df = 57) = 1.76, *p* = 0.084

* This cell had an expected count of less than 5; ** Removed for analysis due to insufficient sample size.

**Table 2 ijerph-19-02255-t002:** Participant well-being, mental health symptoms, and psycho-social characteristics.

	Total Sample (n = 58)	Residents (n = 43)	PA Students (n = 15)	Test Statistic (Degrees of Freedom), *p*-Value
Sleep Disturbance	20.74 (SD = 5.70)	20.07 (SD = 5.36)	22.67 (SD = 6.38)	T(df = 56) = 1.54, *p* = 0.13
Sleep Categories				X^2^(df = 1) = 1.12, *p* = 0.29
None to Slight	41 (70.7%)	32 (74.4%)	9 (60.0%)	
Sleep Disturbance Present	17 (29.3%)	11 (25.6%)	6 (40.0%) *	
Mild	14 (24.1%)	10 (23.3%)	4 (26.7%)	
Moderate	3 (5.2%)	1 (2.3%)	2 (13.3%)	
Severe	0 (0.0%)	0 (0.0%)	0 (0.0%)	
Loneliness	35.77 (SD = 9.81)	36.44 (SD = 9.57)	33.87 (SD = 10.57)	T(df = 56) = −0.87, *p* = 0.39
Depression	3.43 (SD = 3.21)	3.86 (SD = 3.54)	2.20 (SD = 1.57)	T(df = 52.59) = −2.46, *p* = 0.02
Depression Categories				X^2^(df = 1) = 2.96, *p* = 0.09
Normal	40 (69.0%)	27 (62.8%)	13 (86.7%)	
Depression Symptoms Present	18 (31.0%)	16 (37.2%)	2 (13.3%) *	
Mild	9 (15.5%)	7 (16.3%)	2 (13.3%)	
Moderate	8 (13.8%)	8 (18.6%)	0 (0.0%)	
Severe	0 (0.0%)	0 (0.0%)	0 (0.0%)	
Extremely Severe	1 (1.7%)	1 (2.3%)	0 (0.0%)	
Anxiety	3.17 (SD = 3.29)	2.91 (SD = 2.29)	3.93 (SD = 2.60)	T(df = 56) = 1.44, *p* = 0.15
Anxiety Categories				X^2^(df = 1) = 1.88, *p* = 0.17
Normal	32 (55.2%)	26 (60.5%)	6 (40.0%)	
Anxiety Symptoms Present	26 (44.8%)	17 (39.5%)	9 (60.0%)	
Mild	16 (27.6%)	12 (27.9%)	4 (26.7%)	
Moderate	8 (13.8%)	4 (9.3%)	4 (26.7%)	
Severe	2 (3.4%)	1 (2.3%)	1 (6.7%)	
Extremely Severe	0 (0.0%)	0 (0.0%)	0 (0.0%)	
Stress	6.33 (SD = 3.34)	6.58 (SD = 3.58)	5.60 (SD = 2.47)	T(df = 56) = −0.98, *p* = 0.33
Stress Categories				X^2^(df = 1) = 2.96, *p* = 0.09
Normal	40 (69.0%)	27 (62.8%)	13 (86.7%)	
Stress Symptoms Present	18 (31.0%)	16 (37.2%)	2 (13.3%) *	
Mild	8 (13.8%)	7 (16.3%)	1 (6.7%)	
Moderate	7 (12.1%)	6 (14.0%)	1 (6.7%)	
Severe	3 (5.2%)	3 (7.0%)	0 (0.0%)	
Extremely Severe	0 (0.0%)	0 (0.0%)	0 (0.0%)	
Emotional Well-Being	11.36 (SD = 2.48)	11.05 (SD = 2.65)	12.27 (SD = 1.71)	T(df = 1) = 1.66, *p* = 0.10
Social Well-Being	14.48 (SD = 4.87)	14.77 (SD = 5.03)	13.67 (SD = 4.42)	T(df = 56) = −0.75, *p* = 0.46
Psychological Well-Being	22.50 (SD = 4.95)	22.35 (SD = 5.35)	22.93 (SD = 3.69)	T(df = 56) = 0.39, *p* = 0.70
Mental Health				X^2^(df = 1) = 0.19, *p* = 0.66
Flourishing	32 (55.2%)	23 (53.5%)	9 (60.0%)	
Non- Flourishing	26 (44.8%)	20 (46.5%)	6 (40.0%)	
Moderate	25 (43.1%)	19 (44.2%)	6 (40.0%)	
Languishing	1 (1.7%)	1 (2.3%)	0 (0.0%)	

* This cell had an expected count of less than 5.

**Table 3 ijerph-19-02255-t003:** Bivariate analysis of participant characteristics and well-being, mental health symptoms, and psycho-social characteristics.

	Sleep Disturbance r (*p*-Value)	Loneliness r (*p*-Value)	Depression r (*p*-Value)	Anxiety r (*p*-Value)	Stress r (*p*-Value)	Emotional Well-Being r (*p*-Value)	Social Well-Being r (*p*-Value)	Psychological Well-Being r (*p*-Value)
Age	0.06 (0.64)	−0.04 (0.79)	0.21 (0.12)	−0.03 (0.83)	0.03 (0.80)	−0.25 (0.06)	0.01 (0.94)	0.06 (0.64)
Sex	0.006 (0.96)	−0.11 (0.41)	0.08 (0.54)	0.09 (0.53)	0.12 (0.39)	−0.05 (0.71)	0.08 (0.54)	0.01 (0.97)
Relationship Status	−0.02 (0.87)	−0.13 (0.32)	0.20 (0.14)	−0.09 (0.49)	0.31 (0.02)	−0.06 (0.63)	−0.03 (0.82)	0.006 (0.97)
Primary Caregiver	0.21 (0.10)	0.26 (0.05)	0.42 (0.001)	0.21 (0.11)	0.18 (0.18)	−0.15 (0.26)	−0.17 (0.21)	−0.20 (0.12)
Race	0.12 (0.39)	0.20 (0.14)	0.18 (0.17)	0.18 (0.19)	0.11 (0.41)	−0.25 (0.06)	−0.09 (0.51)	−0.25 (0.06)
Hispanic or Latino	0.17 (0.23)	−0.07 (0.63)	−0.17 (0.20)	−0.20 (0.14)	−0.13 (0.36)	0.03 (0.84)	−0.01 (0.94)	−0.002 (0.99)
Specialty	−0.20 (0.12)	0.11 (0.43)	0.13 (0.35)	−0.13 (0.32)	0.09 (0.51)	−0.11 (0.40)	0.08 (0.54)	−0.12 (0.38)
No. of Days Sick in Previous 30 Days	0.25 (0.06)	0.28 (0.04)	0.11 (0.43)	0.34 (0.008)	0.24 (0.07)	−0.08 (0.54)	−0.11 (0.43)	−0.17 (0.21)
No. of Times Exercised in Previous 30 Days	−0.07 (0.59)	−0.37 (0.005)	−0.44 (0.001)	−0.21 (0.11)	−0.29 (0.03)	0.46 (<0.001)	0.29 (0.03)	0.32 (0.02)

**Table 4 ijerph-19-02255-t004:** Well-being, mental health symptoms, and psycho-social characteristics.

	Sleep Dist.	Loneliness	Dep	Anxiety	Stress	Emo. WB	Soc. WB	Psych WB
	r (*p*-Value)	r (*p*-Value)	r (*p*-Value)	r (*p*-Value)	r (*p*-Value)	r (*p*-Value)	r (*p*-Value)	r (*p*-Value)
Sleep Dist.	-	0.22 (0.10)	0.01 (0.95)	0.39 (0.003)	0.02 (0.91)	0.01 (0.96)	−0.04 (0.76)	−0.12 (0.36)
Loneliness		-	0.57 (<0.001)	0.46 (<0.001)	0.27 (0.04)	−0.59 (<0.001)	−0.52 (<0.001)	−0.58 (<0.001)
Dep			-	0.33 (0.01)	0.56 (<0.001)	−0.68 (<0.001)	−0.38 (0.003)	−0.52 (<0.001)
Anxiety				-	0.39 (0.003)	−0.16 (0.24)	−0.31 (0.02)	−0.35 (0.007)
Stress					-	−0.42 (0.001)	−0.44 (0.001)	−0.56 (<0.001)
Emo WB						-	0.50 (<0.001)	0.66 (<0.001)
Soc.							-	0.59 (<0.001)
WB								
Psych WB								-

**Table 5 ijerph-19-02255-t005:** Results from multi-variate logistic regression relating flourishing to participant characteristics, mental health symptoms, and psycho-social characteristics.

	Step 1		Step 2		Step 3	
	AOR (95% CI)	*p*-Value	AOR (95% CI)	*p*-Value	AOR (95% CI)	*p*-Value
Trainee Type (PA Student = 1)	1.08 (0.31–3.78)	0.90	0.81 (0.21–3.08)	0.76	0.57 (0.09–3.60)	0.55
Primary Caregiver (Yes = 1)	0.09 (0.01–0.78)	0.03	0.20 (0.02–2.13)	0.18	0.09 (0.01–1.28)	0.08
# of Days Sick in Previous 30 Days			0.73 (0.48–1.10)	0.13	0.85 (0.44–1.63)	0.63
# of Times Exercised in Previous 30 Days			1.08 (0.97–1.21)	0.18	0.96 (0.79–1.16)	0.66
Loneliness					0.75 (0.61–0.91)	0.003
Depression					1.50 (0.99–2.28)	0.06
Anxiety					1.19 (0.80–1.78)	0.38
Stress					0.65 (0.45–0.94)	0.02
Nagelkerke R^2^	0.16		0.24		0.64	

**Table 6 ijerph-19-02255-t006:** Results from mixed effect linear model analyses relating CTRA gene expression to flourishing and loneliness.

	Model 1 * b (SE) p	Model 2 ** b (SE) p	Model 3 *** b (SE) p
Flourishing (1/0)	−0.173 (0.072) *p* = 0.02	−0.174 (0.087) *p* = 0.06	−0.057 (0.115) *p* = 0.63
Loneliness (z-score)	0.008 (0.003) *p* = 0.02	0.011 (0.005) *p* = 0.03	0.009 (0.006) *p* = 0.17

* Controlling for blood sampling modality (venipuncture vs. DBS). ** Controlling for variables in Model 1 and age, sex, race, current illness symptoms, history of heavy alcohol consumption, and trainee type. *** Controlling for variables in Model 2 and flourishing and loneliness.

## Data Availability

The data presented in this study are available on request from the corresponding author.

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
