# Peer review of "Flourishing in Healthcare Trainees: Psychological Well-Being and the Conserved Transcriptional Response to Adversity"

_ijerph, 2022, doi:10.3390/ijerph19042255_

Round 1
Reviewer 1 Report
This study investigates the demographic and psychosocial correlates of well-being among medical trainees (residents and physician assistant (PA) trainees). In addition to self-report assessments, the study uses RNA sequencing to analyze blood sample RNA profiles. By examining the sense of well-being and fulfillment of healthcare professionals, primarily trainees, the authors provide a resource for considering policies and interventions to strengthen a sustainable healthcare system.
This study provides an essential resource for maintaining and sustaining the current and future healthcare system.
Minor Comments
1)
The research methods in this study are a self-administered questionnaire and the analysis of RNA profiles using blood samples. The results of these analyses are accurately reported in the results section.
In my opinion, the method of data analysis is statistically correct.
In particular, the item "3.3.1 Research question 1: What are the associations among mental health symptoms and psychosocial characteristics with flourishing?" the results are carefully summarized in Table 5.
However, the item "3.3.2. Research Question 2: Are mental health symptoms, psychosocial characteristics, or flourishing associated with pro-inflammatory gene expression?" although the results of a comprehensive analysis are presented, the expression format is summarized in text only. We believe that the relationship between pro-inflammatory gene expression and various items is essential and must be presented. I think that the purpose of this study can be made more evident by showing the results of the analysis in a concise tabular format.
2)
The number of subjects in this study is 59 (table 1), but there is no mention of deletion of missing data or presence of missing data.
(In the "participants" section, it is stated that "There were no exclusion criteria for cohorts.")
If possible, the status of subject selection should be clearly indicated through a flowchart of the study design.
Reviewer 2 Report
For the authors’ guidance my evaluation and some constructive remarks that would help to improve the paper’s quality are included below:
(1) General Comments & Strength of the Paper
General Comments: There are some interesting and important insights in this article. In more detail, these are the main points that could be improved:
1. Introduction
While the introduction provides an overview of some key themes, it remains unclear how these used for this specific piece of work. A few terminologies used in the article need more unpacking at the start, particularly for those approaching the article for the theory and concepts.
Research Questions: The authors specified two research questions: RQ1: What are the associations among mental health symptoms and psychosocial characteristics with flourishing? RQ2: Are mental health symptoms, psychosocial characteristics, or flourishing associated with pro-inflammatory gene expression? Several follow-up questions can be included in the text. In case when research questions are identified, then the authors can attract the attention of readers who are able to conceive the framework of investigation. I recommend the authors to come up with additional follow-up questions, and give precise responses of why the research is interesting and relevant to the field.
Conceptual and Methodological Clarity: The concepts and descriptors used throughout would benefit from more clarity.
2. Literature review
The key arguments would benefit strongly from further fleshing out.
3. Methodology
I recommend the authors clarify their methodology in detail (e.g., research paradigm, research design, research tools, and so on), making sure that their planned methods/research tools are fully detailed. They ought to give attention to justifying their chosen methodology in terms of demonstrating applicability, adjustment, and usefulness in the paper.
Strength: The study indicates the demographic and psychosocial correlations of well-being among healthcare trainees and evaluated well-being’s association with the CTRA. The authors have successfully linked up different approaches and insights. The conceptualization has enriched a substantial depth and theoretical model. The state of the art was adequately substantiated, and the research engaged sufficiently with existing literature.
Originality: I have noticed that there are several passages throughout the paper where phrases and sentences that were taken verbatim from other sources are only referenced with a parenthetical citation when they should also be put in quotation marks. I have detected a 23% iThenticate Similarity Index Analysis Score excluding quotes and bibliography (Please see Attachment for the iThenticate Similarity Index Analysis Report). An international standard for peer-reviewed journals tolerates below 15%.
Sophistication of the Argument
Supportive statement: The topic area is problematised, the discussion has an obvious structure, moving from a general to a more focused theme(s), ideas are clearly/fully developed, and circular reasoning is not used. The conceptualisation has enriched a substantial depth and theoretical model by the application of empirical data. The main ideas presented by the authors are obvious/intelligible and presented in a logical, easy-to-follow manner; the main themes are well-structured and summarised; ideas/insights are not “out-of-the-blue” i.e., they develop as a result of the discussion.
Critique: The article largely lacks precision and at times clarity. The argument is not always clear. Moreover, various statements are postulated without clear argumentation and the connection between the points made remains at times obscure. The theoretical and methodological basis for some of the evaluative statements should at time be made clearer.
4. Concluding Remarks
I appreciate there has been a lot of reading and ground covered. The study may have a stronger focus, compelling argument and discussion, and an indication of why the paper holds value to the readership of the MDPI - IJERPH. I recommend the authors reconsider the approach adopted here; think about the research questions they wish to examine; make sure the literature review is a lot more cohesive, and make sure the link between the research questions and empirical results is a lot “tighter” than presented herewith.
In the light of the above, I recommend revisions but hope this article proceeds to publication thereafter. The authors are on the cusp of a remarkably interesting argument and need to probe and deepen it. Good luck!
